# Alterations in children's sub-dominant gut microbiota by HIV infection and anti-retroviral therapy

**Quynh Thi Nguyen[1], Azumi Ishizaki[1], Xiuqiong Bi[1], Kazunori Matsuda[2], Lam Van Nguyen[3], Hung Viet Pham[3], Chung Thi Thu Phan[3], Thuy Thi Bich Phung[3], Tuyen Thi Thu Ngo[3], An Van Nguyen[3], Dung Thi Khanh Khu[3], Hiroshi Ichimura[1]\***

**1** Department of Viral infection and International Health, Graduate school of Medical Science, Kanazawa University, Kanazawa, Japan, **2** Yakult Central Institute, Tokyo, Japan, **3** Vietnam National Children's Hospital, Hanoi, Viet Nam

☯ These authors contributed equally to this work.

\* ichimura@med.kanazawa-u.ac.jp

## Abstract

### Objective

We investigated the impact of human immunodeficiency virus (HIV) infection and anti-retroviral therapy (ART) on the gut microbiota of children.

### Design

This cross-sectional study investigated the gut microbiota of children with and without HIV.

### Methods

We collected fecal samples from 59 children with HIV (29 treated with ART [ART(+)] and 30 without ART [HIV(+)]) and 20 children without HIV [HIV(−)] in Vietnam. We performed quantitative RT-PCR to detect 14 representative intestinal bacteria targeting 16S/23S rRNA molecules. We also collected the blood samples for immunological analyses.

### Results

In spearman's correlation analyses, no significant correlation between the number of dominant bacteria and age was found among children in the HIV(−) group. However, the number of sub-dominant bacteria, including *Streptococcus*, *Enterococcus*, and *Enterobacteriaceae*, positively correlated with age in the HIV(−) group, but not in the HIV(+) group. In the HIV(+) group, *Clostridium coccoides* group positively associated with the CD4$^+$ cell count and its subsets. In the ART(+) group, *Staphylococcus* and *C. perfringens* positively correlated with CD4$^+$ cells and their subsets and negatively with activated CD8$^+$ cells. *C. coccoides* group and *Bacteroides fragilis* group were associated with regulatory T-cell counts. In multiple linear regression analyses, ART duration was independently associated with the number of *C. perfringens*, and Th17 cell count with the number of *Staphylococcus* in the ART(+) group.

**Data Availability Statement:** All relevant data are within the manuscript and its Supporting information files.

**Funding:** This study was supported, in part, by the Ministry of Education, Culture, Sports, Science, and Technology (MEXT) in Japan (the Program of Founding Research Centers for Emerging and Reemerging Infectious Disease; grant number 23406013, https://www.mext.go.jp/en/publication/whitepaper/title03/detail03/sdetail03/sdetail03/1372928.htm) and the Kanazawa University President Strategic Research Fund (https://www.kanazawa-u.ac.jp/research_bulletin/contact.html). The funders had no role in study design, data collection and analysis, decision to publish, or preparation of the manuscript.

**Competing interests:** The authors have declared that no competing interests exist.

## Conclusions

HIV infection and ART may influence sub-dominant gut bacteria, directly or indirectly, in association with immune status in children with HIV.

## Introduction

The gut microbiota comprises approximately 100 trillion microbes from more than 1000 bacterial species [1, 2]. The gut microbiota plays a major role in nutrient absorption, food metabolism, intestinal barrier protection from pathogens, and the modulation of gut immune function [3–5]. Although the composition of the gut microbiota may be influenced by age, diet, genetics, and geography, four phyla (i.e., Firmicutes, Bacteroidetes, Actinobacteria, and Proteobacteria) are dominant and stable in healthy individuals [4, 6, 7]. CD4+ T cells and their subsets, such as type 1 helper T cells (Th1), Th2, Th17, and regulatory T (Treg) cells, have been associated with the gut microbiota, and their interactions are associated with various diseases, such as inflammatory bowel disease, rheumatoid arthritis, and cancer [8, 9].

Gut-associated lymphoid tissue (GALT) is the largest replication site, and it serves as a reservoir of human immunodeficiency virus type 1 (HIV) [10–12]. Progressive HIV infections are characterized by a depletion of CD4+ T cells in the GALT, followed by microbial translocation, gut microbiota dysbiosis, and chronic immune activation [11, 13–17]. Despite the sustained viral suppression and immune recovery provided by anti-retroviral therapy (ART), the imbalance in gut microbiota is, at best, only partially restored for a long time after initiating ART in adults [15, 16, 18].

In the gut microbiota of healthy children, the dominant phyla are Bacteroidetes and Actinobacteria, particularly the *Bifidobacterium* genus of Actinobacteria. These bacteria have a functional composition similar to that of healthy adults [7, 19–22]. A few studies from Africa and India have shown reduced bacterial diversity in the gut microbiota of children with HIV and children treated with ART compared to the microbiota of children without HIV [23–25]. However, no consensus exists on whether ART in children with HIV may restore the gut microbiota to the state observed in children without HIV [23–25]. The impacts of HIV and ART on the gut microbiota in children remain poorly understood.

In Vietnam, no study has focused on understanding the gut microbiota in children with HIV. Therefore, the current study aimed to investigate the impact of HIV infection and ART on the gut microbiota among children in Vietnam.

## Methods

### Study population

This non-randomized, cross-sectional study was conducted at the Vietnam National Children's Hospital (VNCH) in Hanoi, Vietnam, in 2012 [26, 27]. Children with HIV who did not start ART [HIV(+) group], children with HIV who received ART [ART(+) group], and children without HIV infection [HIV(−) group, control] were recruited.

The inclusion criteria for children with HIV were followed at the VNCH and ≥2 years old. Exclusion criteria were progression of HIV to acquired immunodeficiency syndrome (AIDS), treatment with anything that may influence the immune system, any antibiotics except cotrimoxazole, hospitalization in the prior 8 weeks, or symptoms of gastrointestinal infection, such as diarrhea, nausea, and vomiting, with fever, at the time of recruitment. The children in the

ART(+) group resided at an orphanage near Hanoi. The children in the HIV(+) group resided at their own homes. The children in the HIV(−) group resided at a different orphanage in Hanoi [26, 27].

## Collection and preparation of fecal samples

Immediately after defecation, fecal samples were collected in a plastic container (Sarstedt AG & Co., Nümbrecht, Germany), kept at 4˚C, and transferred to the laboratory using containers maintained at 4˚C. At the laboratory, the fecal samples were weighed, suspended in RNA-stabilizing solution (RNA*later*; Ambion, Inc., Austin, TX, USA), and homogenized (20 mg of feces/mL). The fecal homogenate (200 μL) was added to 1 mL of Dulbecco's Phospahte Buffered Saline (Nissui Pharmaceutical Co., Ltd., Tokyo, Japan). After centrifuging the mixture at 12,000 × *g* for 5 min, the pellet was stored at −80˚C until used for RNA extraction. The whole process was completed within 24 hours after defecation [28].

## Quantification of bacteria in human feces by RT-qPCR

Total RNA extraction and subsequent reverse-transcription and quantitative polymerase chain reaction (RT-qPCR) assays were performed using the methods described by Matsuda et al. [29, 30]. Briefly, 4 mg of feces were subjected to total RNA extraction, and each extracted RNA sample was serially diluted 10-fold. Three serial dilutions of the extracted RNA samples (corresponding to 1/400, 1/4,000, and 1/40,000 of the extracted RNA) were subjected to RT-qPCR with specific primer sets that targeted the 16S or 23S rRNA of the 14 representative intestinal bacteria in four main phyla, including: Firmicutes, such as *Clostridium coccoides* group, *C. leptum* subgroup, *C. difficile*, *C. perfringens*, *Lactobacillus* spp., *Streptococcus*, *Enterococcus*, and *Staphylococcus*; Actinobacteria, such as *Bifidobacterium* and *Atopobium* cluster; Bacteroidetes, such as *Bacteroides fragilis* group and *Prevotella*; and Proteobacteria, such as *Enterobacteriaceae* and *Pseudomonas* [29–32]. The counts of *Lactobacillus* spp. obtained with RT-qPCR were expressed as the sum of six *Lactobacillus* subgroups and two species. In the same experiment, a standard curve was generated with the RT-qPCR data (by threshold cycle: $C_T$ value) and the cell counts (by DAPI staining) of the dilution series of total RNA from the standard strain for each bacterial target. The $C_T$ values from fecal samples were normalized to the standard curve to obtain the bacterial count per gram wet weight of feces.

In addition, the individual bacteria in the fecal microbiota are present at different microbial cell counts. Previous reports revealed that the average total bacterial count is approximately $10^{11}$ cells/g of feces [29, 31]. We regarded the threshold for dominance in abundance at 1.0% of the total bacterial count, and the threshold in counts was set at $10^9$ cells/g [6, 33–35].

## Laboratory methods

White blood cell (WBC) counts, WBC differentiation, hemoglobin level (Beckman Coulter, Lh 780, USA), total cholesterol level, and fasting blood sugar (Olympus AU640, Japan) were measured at the clinical laboratory of VNCH. Plasma HIV viral loads were measured as described previously [26].

## Immunological analysis

Immunological investigations were performed with blood samples as described previously [26]. Briefly, whole blood samples were stained with a combination of monoclonal antibodies to detect cell surface molecules within 6 hours after collection and analyzed using a JSAN flow cytometer (Bay Bioscience, Kobe, Japan). The obtained data were analyzed by Flowjo V.7.5.5

(FLOWJO, OR, USA). We defined CD38$^+$HLA-DR$^+$CD8$^+$ cells as activated CD8$^+$ cells, CXCR3$^+$CCR6$^-$CD25$^{low}$CD4$^+$ cells as Th1, CXCR3$^-$CCR6$^-$CD25$^{low}$CD4$^+$ cells as Th2, CXCR3$^-$CCR6$^+$CD25$^{low}$CD4$^+$ cells as Th17 cells, and CD25$^{high}$CD4$^+$ cells as Treg cells [36–38]. The gating strategy for cell staining analysis by flow cytometry is shown as S1 Fig. The absolute cell count was calculated as WBC count × percentage of lymphocytes × percentage of target cells obtained by flow cytometry.

## Statistical analysis

Statistical analyses were performed using SPSS version 25 (IBM SPSS Statistics, USA) and R version 3.6.2 [39]. The chi-squared test or Fisher's exact test was performed to assess the differences in bacterial detection rates. The gut microbiota patterns were presented as biplots with the principal component analysis (PCA) using the *prcomp* function from the ggbiplot package in R. The number of bacteria was compared between the groups using the Mann-Whitney *U* test. Spearman's rank test was used to analyze the pairwise correlations between bacteria and possibly related factors, such as age, ART duration, CD4$^+$ cells and their subsets, CD8$^+$ cells, the proportion of activated CD8$^+$ cells, and the use of cotrimoxazole. The correlations were visualized as a heatmap using the *corrplot* function in R. Simple linear regression was used to assess the linear relationship of the significantly correlated pairs. The significant relationship was confirmed in multiple linear stepwise regression analysis. CD4$^+$ cells were not included in the multiple linear regression analysis due to the multicollinearity with their subsets. $P < 0.05$ was considered significant.

## Study approval

This study was carried out according to the World Medical Association's Declaration of Helsinki, the Japanese Ethics Guidelines for Human Genome/Gene Analysis Research, and the Vietnamese Ethics Guidelines. The protocol was approved by the Ethics Committee of Kanazawa University [2011–080 (5775)] and the Ethics Committee of the VNCH (09/2012/BVNTWW-HD3), Hanoi, Vietnam. Each child's parents or guardians were informed, and written consent was obtained for all participants. This study is registered at UMIN-CTR: UMIN000015044.

## Results

### Recruitment and characteristics of the study population

Approximately 500 children with HIV were followed at the VNCH in Hanoi, Vietnam, in 2012, and 40 of them did not start ART according to the Guidelines of the Ministry of Health in Vietnam (No. 3003/QĐ-BYT dated 19/08/2009) [40]. We invited all 40 of the children who did not receive ART, 30 of whom consented to participate in this study [13 females and 17 males; median age 5.9 years, range 2.0–8.8 years; HIV(+) group]. We tried to recruit age- and gender-matched children with HIV who were on ART [n = 29: 12 females and 17 males; median age 6.1 years, range 3.6–8.6 years; median duration of ART: 3.5 years, range 0.8–5.8 years; ART(+) group] and children without HIV as a control [n = 20, 8 females and 12 males; median age 4.1, range 2.0–8.3 years; HIV(–) group], though we could only recruit a smaller number of children without HIV who were 2 years younger than the HIV(+) and ART(+) groups ($P = 0.048$ and $P = 0.009$, respectively). Their detailed demographic characteristics and immune status are provided in Table 1 and elsewhere [26, 27].

The 29 children in the ART(+) group were all treated with two nucleoside reverse transcriptase inhibitors (NRTIs); 25 were also treated with one non-nucleoside reverse transcriptase

**Table 1. Characteristics and immune status of each study group [25].**

| Characteristic | HIV(−) (n = 20) | HIV(+) (n = 30) | ART(+) (n = 29) | P-values | | |
|---|---|---|---|---|---|---|
| | | | | HIV(+) vs. HIV(−) | ART(+) vs. HIV(−) | HIV(+) vs. ART(+) |
| Age (years) | 4.1 (2.0–8.3) | 5.9 (2.0–8.8) | 6.1 (3.6–8.6) | 0.048 | 0.009 | 0.44 |
| Gender, F/M | 8/12 | 13/17 | 12/17 | 0.82 | 0.92 | 0.89 |
| Height (cm) | 110 (80–130) | 107.5 (77–129.5) | 110 (90–130) | 0.88 | 0.42 | 0.41 |
| Body weight (kg) | 16 (9–35) | 17.3 (10–27) | 19.8 (12.0–32.8) | 0.47 | 0.14 | 0.14 |
| Hemoglobin (g/L) | 130.5 (114–141) | 121.5 (83–139) | 129 (104–157) | 0.001 | 0.74 | 0.001 |
| Total cholesterol (mmol/L) | 3.9 (3.2–5.0) | 2.8 (1.8–4.3) | 3.8 (2.8–5.3) | <0.001 | 0.57 | <0.001 |
| Fasting blood sugar (mmol/L) | 4.9 (4.2–5.3) | 3.8 (2.6–8.3) | 5.1 (3.5–6.0) | < 0.001 | 0.03 | <0.001 |
| WHO clinical stage, 2/1 | | 1/29 | 5/24 | | | 0.10 |
| ART duration (years) | | | 3.5 (0.8–5.8) | | | |
| Age of ART initiation (years) | | | 2.7 (0.4–6.9) | | | |
| Viral load ($\log_{10}$ copies/μL) | | 5.0 (3.2–6.5) | 3.6 (2.4–4.4)* | | | <0.001 |
| CD4+ cell count (cells/μL) | 1050 (693–2688) | 691 (97–1784) | 894 (244–1711) | 0.003 | 0.018 | 0.43 |
| Th1 count (cells/μL) | 136 (74–220) | 78 (25–227) | 147 (49–211) | 0.004 | 0.61 | 0.003 |
| Th2 count (cells/μL) | 822 (413–2196) | 532 (63–1375) | 553 (119–1369) | 0.017 | 0.009 | 0.98 |
| Th17 count (cells/μL) | 109 (51–192) | 45 (6–116) | 58 (23–144) | <0.001 | <0.001 | 0.02 |
| Treg count (cells/μL) | 48 (16–94) | 14 (0–133) | 30 (9–71) | <0.001 | 0.004 | <0.001 |
| CD8+ cell count (cells/μL) | 1101 (634–2874) | 1417 (470–3127) | 1212 (769–2064) | 0.24 | 0.29 | 0.64 |
| Activated CD8+ cells (%) | 12.9 (5.8–38.6) | 28.3 (12.2–53.3) | 10.2 (5.0–27.7) | <0.001 | 0.33 | <0.001 |
| CD4+/CD8+ ratio | 1.03 (0.45–2.34) | 0.49 (0.06–1.18) | 0.66 (0.20–1.42) | <0.001 | 0.001 | 0.19 |

Values are given as the median (range) or the number of patients. F: female: M: male; HIV(+): children with HIV and not treated with ART; ART(+): children with HIV and treated with ART; HIV(−): children not infected with HIV. P-values are based on the Mann-Whitney U test, except for the gender and WHO clinical stage comparison, which is based on the chi-squared test or Fisher's exact test.

*Viral load was undetectable in 22 children in the ART(+) group.

inhibitor and the remaining 4 with one protease inhibitor (PI): 8 received zidovudine/lamivudine/nevirapine; 7 received stavudine/lamivudine/nevirapine; 6 received zidovudine/lamivudine/efavirenz; 4 received stavudine/lamivudine/efavirenz; 2 received zidovudine/lamivudine/lopinavir boosted with ritonavir; 1 received abacavir/lamivudine/lopinavir boosted with ritonavir; and 1 received abacavir/didanosine/lopinavir boosted with ritonavir. Fifteen children in the HIV(+) group and nine children in the ART(+) group received cotrimoxazole according to the Guidelines of the Ministry of Health in Vietnam (No. 3003/QĐ-BYT dated 19/08/2009) [40].

## Gut microbiota profiles

The dominant bacteria in the gut microbiota ($\geq 10^9$ cells/g of feces) included *C. coccoides* group, *C. leptum* subgroup, *Bifidobacterium*, *Atopobium* cluster, *B. fragilis* group, and *Prevotella*. The sub-dominant gut microbiota ($<10^9$ cells/g) included *C. difficile*, *C. perfringens*, *Streptococcus*, *Enterobacteriaceae*, *Lactobacillus* spp., *Enterococcus*, *Staphylococcus*, and *Pseudomonas* (S1 Table). Due to the low detection frequencies of *C. difficile* and *Pseudomonas* (3.4% to 20% in all groups), these two bacteria were not included in further analyses (S2 Table).

PCA revealed that the HIV(−) and HIV(+) groups had similar gut microbiota structures. The gut microbiota structure of the ART(+) group was different from the other groups and characterized by the abundance of *Atopobium* cluster, *Bifidobacterium*, *Prevotella*, and *Lactobacillus* spp. (Fig 1).

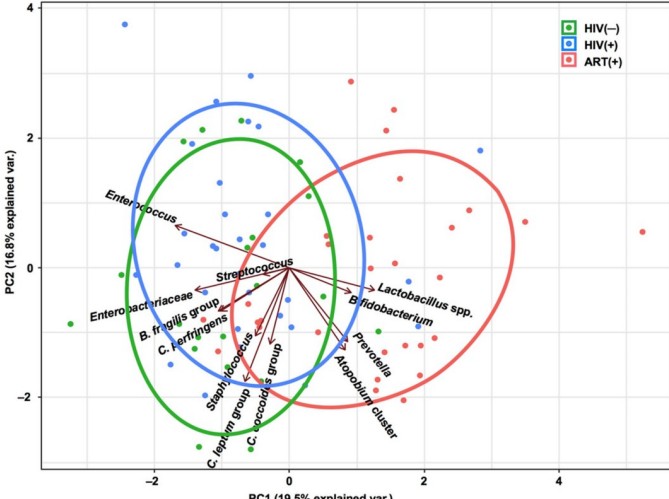

**Fig 1. Principal component analysis based on the overall structure of the gut microbiota in three groups of children.** Each data point represents an individual sample. Ellipses represent the 95% confidence level. Treatment groups are color-coded: green, HIV(−); blue, HIV(+); and red, ART(+). Arrows indicate characteristic vectors of the 12 bacterial factors.

The number of bacteria in each group is shown in Fig 2. In the HIV(+) group, the numbers of *C. perfringens* and *Atopobium* cluster were significantly lower (*P* = 0.02 and *P* = 0.048, respectively) and the number of *Lactobacillus* spp. significantly higher (*P* = 0.02) than in the HIV(−) group. In the ART(+) group, the numbers of *Enterococcus*, *B. fragilis* group, and

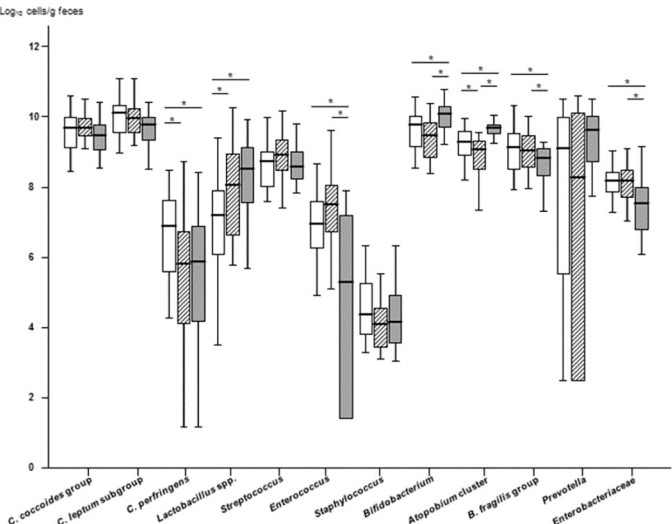

**Fig 2. Box plots showing the abundance of bacteria in the gut microbiota of the three study groups.** Phylum Firmicutes: *Clostridium coccoides* group, *C. leptum* subgroup, *C. perfringens*, *Lactobacillus* spp., *Streptococcus*, *Enterococcus*, and *Staphylococcus*; phylum Actinobacteria: *Bifidobacterium* and *Atopobium* cluster; phylum Bacteroidetes: *Bacteroides fragilis* group and *Prevotella*; and phylum Proteobacteria: *Enterobacteriaceae*. The lines and error bars correspond to the medians ± 95% confidence intervals. White box, HIV(−) group; oblique line box, HIV(+) group; gray box, ART(+) group. *$P < 0.05$, Mann-Whitney *U* test. *C. difficile* and *Pseudomonas* were not included due to the low detection frequencies (3.4% to 20% in all three groups; *C. difficile*, median = 1.15 $\log_{10}$ cells/g feces, and *Pseudomonas*, median = 1.45 $\log_{10}$ cells/g feces, S2 Table).

*Enterobacteriaceae* were significantly lower ($P < 0.001$, $P = 0.04$, $P = 0.002$, respectively) and the numbers of *Bifidobacterium* and *Atopobium* cluster significantly higher (both $P < 0.001$) than in the HIV(+) group (Fig 2 and S1 Table).

## Association between age and gut microbiota

In the HIV(−) group, but not in the HIV(+) group, the numbers of *Streptococcus*, *Enterococcus*, and *Enterobacteriaceae* positively correlated with age (Rho = 0.59, $P = 0.006$; Rho = 0.61, $P = 0.005$; and Rho = 0.57, $P = 0.008$, respectively; Fig 3 and S3 Table). The number of *Staphylococcus* inversely correlated with age in the HIV(+) group (Rho = −0.47, $P = 0.009$).

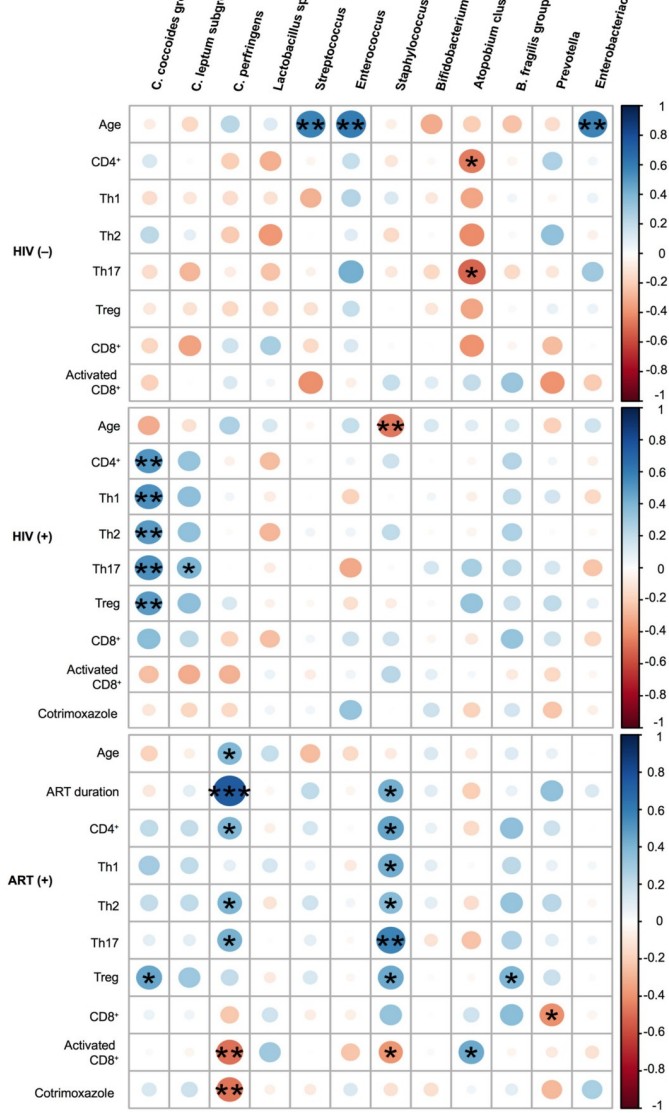

**Fig 3. Heatmap representing the correlation of gut microbiota with age, anti-retroviral therapy (ART) duration, immune status, and use of cotrimoxazole.** Blue shading indicates a positive association and red shading a negative association. The scale indicates the strengths of associations. *C. difficile* and *Pseudomonas* were not included due to the low detection frequencies (3.4% to 20% in all three groups: *C. difficile*, median = $\log_{10}$ 1.15 cells/g feces, and *Pseudomonas*, median = $\log_{10}$ 1.45 cells/g feces, S2 Table). The color intensity and size of the circles are proportional to the correlation coefficients. *$P < 0.05$, **$P < 0.01$, ***$P < 0.001$, based on Spearman's rank-test.

## Association between gut microbiota and immune status

In the HIV(−) group, the number of *Atopobium* cluster inversely correlated with CD4$^+$ cell and Th17 counts (Rho = −0.46, $P$ = 0.04 and Rho = −0.51, $P$ = 0.02, respectively). In the HIV(+) group, the number of *C. coccoides* group significantly correlated with the CD4$^+$ cell count and its subsets (CD4$^+$ cells: Rho = 0.51, $P$ = 0.004; Th1: Rho = 0.54, $P$ = 0.002; Th2: Rho = 0.50, $P$ = 0.005; Th17: Rho = 0.54, $P$ = 0.002; and Treg: Rho = 0.49, $P$ = 0.006).

In the ART(+) group, the number of *Staphylococcus* significantly correlated with the CD4$^+$ cell count and its subsets (CD4$^+$ cells: Rho = 0.46, $P$ = 0.01; Th1: Rho = 0.44, $P$ = 0.02; Th2: Rho = 0.37, $P$ = 0.047; Th17: Rho = 0.58, $P$ = 0.001; Treg: Rho = 0.45, $P$ = 0.02), the percentage of activated CD8$^+$ cells (Rho = −0.39, $P$ = 0.04), and the ART duration (Rho = 0.42, $P$ = 0.02). The number of *C. perfringens* significantly correlated with age (Rho = 0.39, $P$ = 0.03), CD4$^+$ cell count (Rho = 0.39, $P$ = 0.04), Th2 count (Rho = 0.40, $P$ = 0.03), Th17 count (Rho = 0.42, $P$ = 0.03), and percentage of activated CD8$^+$ cells (Rho = −0.49, $P$ = 0.01), and most strongly with the ART duration (Rho = 0.75, $P$ < 0.001). The *C. coccoides* group and *B. fragilis* group were associated with Treg cell count (Rho = 0.45, $P$ = 0.01 and Rho = 0.40, $P$ = 0.03, respectively). *Prevotella* was negatively associated with the CD8$^+$ cell count (Rho = −0.41, $P$ = 0.03; Fig 3 and S3 Table).

## Impact of cotrimoxazole on the gut microbiota of children with HIV

Fifteen (50.0%) of the 30 children in the HIV(+) group and 9 (31%) of the 29 children in the ART(+) group received cotrimoxazole. Cotrimoxazole treatment did not significantly affect the gut microbiota profile in the HIV(+) group. However, in the ART(+) group, the number of *C. perfringens* was significantly lower among children treated with cotrimoxazole than those not treated with cotrimoxazole [with cotrimoxazole: 3.4 (2.2–5.4) vs. without cotrimoxazole: 6.2 (4.8–7.5), $P$ = 0.01; S4 Table].

## Independent predictors of the gut microbiota in children with HIV

The multiple linear regression analyses including age, ART duration, immune status, and use of cotrimoxazole showed that the ART duration was independently associated with the number of *C. perfringens* (Beta coefficient = 0.726, $P$ < 0.001), and Th17 count with the number of *Staphylococcus* (Beta coefficient = 0.428, $P$ = 0.02) in the ART(+) group (Tables 2 and 3). The linear regression analysis for *C. coccoides* group in the HIV(+) group showed no significant association (S5 Table).

**Table 2. Linear regression analysis of *Clostridium perfringens* with age, ART duration, immune status, and use of cotrimoxazole in the ART(+) group.**

| Variable | Unadjusted linear regression | | | Adjusted linear regression | | |
|---|---|---|---|---|---|---|
| | **Beta** | **SE** | ***P*-value** | **Beta** | **SE** | ***P*-value** |
| Age (years) | 0.323 | 0.261 | 0.09 | | | |
| ART duration (years) | 0.726 | 0.166 | <**0.001** | 0.726 | 0.166 | <**0.001** |
| Th2 count | 0.418 | 0.001 | **0.024** | | | |
| Th17 count | 0.371 | 0.015 | **0.048** | | | |
| Activated CD8$^+$ cells (%) | -0.484 | 0.068 | **0.008** | | | |
| Cotrimoxazole use (yes vs. no) | -0.446 | 0.809 | **0.015** | | | |

ART: anti-retroviral therapy; Beta: regression coefficient; SE: standard error.

The factors with $P$ < 0.05 in the simple linear regression analysis were included in the stepwise multiple linear regression analysis. $P$-values in bold are significant.

**Table 3. Linear regression analysis of *Staphylococcus* with ART duration and immune status in the ART(+) group.**

| Variable | Unadjusted linear regression | | | Adjusted linear regression | | |
|---|---|---|---|---|---|---|
| | **Beta** | **SE** | ***P*-value** | **Beta** | **SE** | ***P*-value** |
| ART duration (years) | 0.370 | 0.109 | **0.048** | | | |
| Th1 count | 0.309 | 0.005 | 0.103 | | | |
| Th2 count | 0.279 | 0.001 | 0.142 | | | |
| Th17 count | 0.428 | 0.007 | **0.020** | 0.428 | 0.007 | **0.020** |
| Treg count | 0.244 | 0.012 | 0.202 | | | |
| Activated CD8$^+$ cells (%) | -0.305 | 0.036 | 0.108 | | | |

ART: anti-retroviral therapy; Treg: regulatory T cells; Beta: regression coefficient; SE: standard error. The factors with $P < 0.05$ in the simple linear regression analysis were included in the stepwise multiple linear regression analysis. *P*-values in bold are significant.

## Discussion

In the present study, we investigated the impact of HIV infection and ART on the gut microbiota of Vietnamese children using RT-qPCR. We found that several sub-dominant gut bacteria were positively associated with age in children without HIV, but this was not observed in the children with HIV. In addition, *Staphylococcus* negatively correlated with age, i.e. the duration of HIV infection, in the children vertically infected with HIV, and ART duration had an independent positive association with *C. perfringens* and Th17 count with *Staphylococcus* in the HIV-infected children on ART. These findings indicate an impact of HIV infection and ART on the sub-dominant gut microbiota, such as *C. perfringens* and *Staphylococcus*, in children. Our findings highlight the importance of investigating the role of the sub-dominant gut microbiota in the pathogenesis of HIV infection.

To the best of our knowledge, this study is the first to apply RT-qPCR techniques to evaluate the gut microbiota, particularly sub-dominant bacteria, in children with and without HIV. The sum of the six dominant bacterial groups in fecal samples detected by RT-qPCR was previously reported to cover 71.3% of the total intestinal bacterial count estimated by hybridization with a universal probe [29, 41]. This finding ensures the validity of using RT-qPCR methods to identify the main gut microbiota profile in this study, though our results may not be comparable directly to the results of the other studies using NGS, since the RT-qPCR method is not appropriate to calculate the diversity indices and the relative abundance of the selected bacteria. In addition, the RT-qPCR approach can detect and enumerate the gut bacteria at the population level between $10^2$ and $10^{11}$ cells/g of feces, whereas the lower detection limit of next generation sequencing (NGS) methods is $10^7$ to $10^8$ cells/g [33]. The counts of the sub-dominant bacteria, including *Lactobacilli* and potential pathogens, such as *C. perfringens*, were near the detection limit of NGS or lower [33]; thus, we took advantage of RT-qPCR to estimate the counts of these less abundant, but clinically significant, targets.

In this study, the numbers of dominant gut bacteria, including *C. coccoides* group, *C. leptum* subgroup, *Bifidobacterium*, and *Atopobium* cluster, did not correlate with age in the HIV(−) group, i.e., children aged 2 to 8 years. This finding is consistent with previous findings that the gut microbiota of healthy children stabilizes and becomes similar to that of adults at around 2 or 3 years of age [7, 21, 22, 33]. In contrast, several sub-dominant gut bacteria, including *Streptococcus*, *Enterococcus*, and *Enterobacteriaceae*, positively correlated with age in the HIV(−) group. This finding is also consistent with previous findings in Japanese children evaluated using the same RT-qPCR methods [33]. These results suggest that the dominant gut

microbiota may reach stable levels by 2 or 3 years of age, whereas the sub-dominant bacteria may still be in transition in children aged 2 to 8 years.

We found that *C. coccoides* group positively correlated with the CD4$^+$ cell count and its subsets in the HIV(+) group, as well as the Treg cell count in the ART(+) group. The *C. coccoides* spp. are known to promote the expansion and differentiation of Treg cells, which play a central role in regulating gut inflammation through the production of butyrate, a short-chain fatty acid (SCFA), in mice [42, 43]. In the ART(+) group, the *B. fragilis* group positively correlated with the Treg cell count. This finding is consistent with a previous study that found that *B. fragilis* promotes Treg cell function by producing polysaccharide A [44]. Thus, HIV infection and ART may also influence the immune status by changing the levels of gut bacterial metabolites, such as SCFAs [45, 46]. Further studies on bacterial metabolites and their networks in the gut microbiota of children with HIV treated with and without ART may elucidate the underlying mechanisms of immune modulation in HIV infection and ART interventions.

Multiple regression analysis showed a positive association between gut *Staphylococcus* and Th17 count in the ART(+) group, which was shown for the first time. Th17 cells produce interleukin-17, which is important for promoting neutrophil recruitment to clear bacteria and has a specific role in the host defense against *Staphylococcus aureus* skin infection [47]. Thus, it would be interesting to investigate the interaction between Th17 and gut *Staphylococcus* in order to understand the pathophysiology of HIV infection in children who are on ART.

The use of cotrimoxazole reportedly influences some gut bacteria and reduces gut inflammation in children with HIV [48–50]. In the current study, the use of cotrimoxazole was associated only with *C. perfringens* in the ART(+) group. However, in multiple regression analysis, we found that ART duration, but not the use of cotrimoxazole, was independently associated with *C. perfringens*, which is a potentially harmful bacterium [51]. These findings suggest that a novel therapeutic approach, such as ingesting probiotics and/or prebiotics, may be necessary to restore gut microbiota homeostasis in children with HIV who are on ART [52].

In this study, all of the children in the ART(+) group were treated with NRTIs as backbone drugs and only four also received a PI. Therefore, the positive correlation between the number of *C. perfringens* and ART duration may be due to the use of NRTIs, but not PIs, even though PIs are known to lower the diversity of the gut microbiota [53, 54]. Our study is consistent with the previous study that ART, especially NRTI-including regimen, had more suppressive impacts on the composition and the variability of the gut microbiota [55]. Further study is needed to investigate whether NRTIs influence the gut microbiota, directly or indirectly, through the restoration of immune status in children with HIV who are on ART.

This study has some limitations. First, the children in the HIV(−) group were 2 years younger than the other groups. The diets were not controlled among the groups, though the children in the HIV(−) and ART(+) groups who resided at orphanages were provided the same diets. The children in the HIV(+) group who resided in their own homes appeared to have poorer nutritional status than the children in other groups, which could be due to the uncontrolled diet and/or HIV infection [26]. Considering the influence of age and diet factors on the gut microbiota [7, 21, 22, 33, 56–58], we did not focus on comparing the gut microbiota between the groups, but highlighted the factors associated with the gut microbiota in each group. Second, the number of patients recruited in the present study was relatively small, which may limit the significance of our findings. Third, we have not mentioned the HIV-exposure history of the children in the HIV(−) group because we could not confirm the history via documents, though they were reportedly born to mothers with HIV. Machiavelli *et al.* reported that the gut microbiota of HIV-exposed but uninfected children is similar to that of HIV-unexposed and uninfected children at the age of 18 months except in several *taxa* [59]. Therefore, the impact of HIV-exposure history on the gut microbiota in the children without HIV over 2

years of age would be limited. These findings require confirmation in longitudinal studies that compare the gut microbiome between age-matched children with and without HIV, with and without HIV-exposure history, and/or before and after initiating ART to assess the effect of ART on the composition of the gut microbiota in children with HIV.

This study provided new insights into the alterations in the gut microbiota, particularly the sub-dominant groups of bacteria, among children with HIV. Our results suggest that HIV infection and ART influence the sub-dominant gut microbiota, directly or indirectly, in association with the immune status of children with HIV.

## Supporting information

**S1 Fig. The gating strategy for cell staining analysis in flow cytometry.** $CD8^+$ cell activations was defined as the $CD38^+HLA-DR^+$ population. Regulatory T (Treg) cells were defined as $CD25^{high}CD4^+$ cells, Th1 as $CXCR3^+CCR6^-CD25^{low}CD4^+$ cells, Th2 as $CXCR3^-CCR6^-CD25^{low}CD4^+$ cells, and Th17 as $CXCR3^-CCR6^+CD25^{low}CD4^+$ cells. Unstained cells were used for gating controls.
(TIF)

**S1 Table. Number of bacteria in fecal samples from each study group.** Values are the median counts (IQR), based on RT-qPCR, expressed in units of log10 cells/g feces. The Lactobacillus spp. counts were obtained with RT-qPCR and are expressed as the sum of the six subgroups and two species; ART: anti-retroviral therapy; C.: Clostridium; L.: Lactobacillus; B.: Bacteroides. P-values in bold are statistically significant, based on the Man-Whitney U test.
(DOCX)

**S2 Table. Detection frequency of bacteria in fecal samples from each study group.** Values are the detection frequency, defined as the % of samples that harbored detectable microbiota, among all samples in a given group. The Lactobacillus spp. counts were obtained with RT-qPCR and are expressed as the sum of the six subgroups and two species; ART: anti-retroviral therapy; C.: Clostridium; L.: Lactobacillus; B.: Bacteroides; P-values in bold are statistically significant, based on the Chi-square test or Fisher's exact probability test.
(DOCX)

**S3 Table. Association between gut microbiota and age, ART duration, immune status, and use of cotrimoxazole in children.** ART: anti-retroviral therapy; Treg: regulatory T cell; C.: Clostridium; B.: Bacteroides; P-values in bold are statistically significant, based on Spearman's rank correlation analysis.
(DOCX)

**S4 Table. Number of bacteria in fecal samples from the HIV(+) and ART(+) groups stratified by the use of cotrimoxazole.** Values are the median counts (IQR), based on RT-qPCR, expressed in units of $log_{10}$ cells/g feces; ART: anti-retroviral therapy; *C.*: *Clostridium*; *B.*: *Bacteroides*; *P*-values in bold are statistically significant, based on the Mann−Whitney *U* test.
(DOCX)

**S5 Table. Linear regression analysis of *Clostridium coccoides* group with immune status in the HIV(+) group.** ART: anti-retroviral therapy; Treg: regulatory T cells; Beta: regression coefficient; SE: standard error. Stepwise multiple linear regression analysis was not done, since no factor with $P < 0.05$ was found in the simple linear regression analysis.
(DOCX)

## Acknowledgments

We would like to thank the children who participated in this study for their invaluable support through the use of their samples. We would also like to thank Ms. Thuy Thi Thanh Giang and other staff at the Department of Infectious Diseases and Molecular Laboratory of VNCH, who contributed enormously to this work.

## Author Contributions

**Conceptualization:** Azumi Ishizaki, Hiroshi Ichimura.

**Data curation:** Quynh Thi Nguyen, Azumi Ishizaki, Xiuqiong Bi, Kazunori Matsuda, Lam Van Nguyen, Hung Viet Pham, Chung Thi Thu Phan, Tuyen Thi Thu Ngo.

**Formal analysis:** Quynh Thi Nguyen, Azumi Ishizaki, Kazunori Matsuda.

**Funding acquisition:** Hiroshi Ichimura.

**Investigation:** Quynh Thi Nguyen, Azumi Ishizaki, Xiuqiong Bi, Kazunori Matsuda, Hung Viet Pham, Chung Thi Thu Phan.

**Methodology:** Azumi Ishizaki, Xiuqiong Bi, Kazunori Matsuda, Hung Viet Pham, Thuy Thi Bich Phung.

**Project administration:** Tuyen Thi Thu Ngo, Dung Thi Khanh Khu, Hiroshi Ichimura.

**Resources:** Azumi Ishizaki, Lam Van Nguyen, Hung Viet Pham, Chung Thi Thu Phan, Tuyen Thi Thu Ngo, An Van Nguyen, Dung Thi Khanh Khu.

**Software:** Quynh Thi Nguyen, Azumi Ishizaki, Xiuqiong Bi, Kazunori Matsuda, An Van Nguyen.

**Supervision:** Hiroshi Ichimura.

**Validation:** Quynh Thi Nguyen, Xiuqiong Bi, Kazunori Matsuda, Thuy Thi Bich Phung, An Van Nguyen, Hiroshi Ichimura.

**Visualization:** Quynh Thi Nguyen, Azumi Ishizaki, Xiuqiong Bi.

**Writing – original draft:** Quynh Thi Nguyen, Azumi Ishizaki.

**Writing – review & editing:** Xiuqiong Bi, Kazunori Matsuda, Lam Van Nguyen, Hung Viet Pham, Chung Thi Thu Phan, Thuy Thi Bich Phung, Tuyen Thi Thu Ngo, An Van Nguyen, Dung Thi Khanh Khu, Hiroshi Ichimura.

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
