## [Decision Letter · Decision Letter 0]

13 Jul 2021

PONE-D-21-16737

Alterations in children's sub-dominant gut microbiota by HIV infection and anti-retroviral therapy

PLOS ONE

Dear Dr. Ichimura,

Thank you for submitting your manuscript to PLOS ONE. After careful consideration, we feel that it has merit but does not fully meet PLOS ONE’s publication criteria as it currently stands. Therefore, we invite you to submit a revised version of the manuscript that addresses the points raised during the review process.

Please be sure to pay particular attention to the following reviewer concerns: 

Ensure that all relevant literature is cited and the findings in the current study are put into context with the additional references Provide additional justification for the selection of flow cytometry markers and update gating strategy figure Provide justification for the experimental methods selected for the analysis of bacterial communities and include additional analyses on diversity measures and relative abundance analysis Address questions with regards to the impacts of age and ART duration and incorporate into manuscriptAddress all other major and minor reviewer comments/concerns 

If you choose not to submit a revision, please notify us. 

We look forward to receiving your revised manuscript.

Kind regards,

Dr. Jennifer Manuzak

Academic Editor

PLOS ONE

Reviewers' comments:

Reviewer's Responses to Questions

**Comments to the Author**

1. Is the manuscript technically sound, and do the data support the conclusions?

Reviewer #1: Partly

Reviewer #2: Partly

2. Has the statistical analysis been performed appropriately and rigorously? 

Reviewer #1: Yes

Reviewer #2: Yes

3. Have the authors made all data underlying the findings in their manuscript fully available?

Reviewer #1: Yes

Reviewer #2: Yes

4. Is the manuscript presented in an intelligible fashion and written in standard English?

Reviewer #1: Yes

Reviewer #2: Yes

5. Review Comments to the Author

Reviewer #1: This study examined select bacteria via RT-PCR in the feces of children from Vietnam who were HIV- or HIV+ (ART+/-). The authors then looked for associations between the absolute number of bacteria and a variety of other parameters (Age of child, length of time on ART, T cell counts, activation of CD8+ T cells and use of cotrimoxazole) antibiotic.

Major

1. This study is similar to previous work including Flygel et al 2020 Journ Infec Dis and Kaur et al 2018 Sci Rep, which the authors did cite in the introduction. This work is also similar to the studies by Dirajlal-Fargo et al 2020 AIDS (Brazil) and Abange et al 2021 Sci Repo (Cameroon) which was not cited by the authors.

A). Since the current study is so similar to other studies, the authors must provide additional justification for the current study in the introduction.

B). The authors should reference the additional two studies listed above.

C). There is no mention of these comparable studies in the discussion. The current study’s findings must be discussed in the context of existing literature (i.e. what findings were similar between studies, what findings were different and speculation on why there are differences between studies). For example, in the Flygel et al study, children were on ART for at least 6 months at the time of sampling. How does this differ since it is known that the length of time on ART impacts gut microbiome (Imahashi et al 2021)? How did T cell activation differ between the appropriate studies? Does geographical region impact differences between studies?

2. The gating strategy in S1 Figure needs some modification.

A). For the CD8 vs CD4 flow plot, most of the events appear to be out of view on the axis. This flow plot needs to be adjusted so that all events are brought into view.

B). For the CD38 vs HLA-DR flow plot, please provide and isotype control flow plot or FMO controls. The authors may be missing some of the activated CD8 cells based on where the quadrant gate currently sits.

C). It is confusing why the authors have an arrow pointing downwards from the lymphocyte gate towards CD4 when they could have gated from the CD4+ cells found in the CD8 vs CD4 flow plot.

4. The authors need to provide justification for why CD3 was not included in the panel since monocytes and NK cells in the blood can express CD4 and CD8.

Minor

1. Provide some explanation in the discussion as to why the antibiotic cotrimoxazole only effected C. perfringens.

2. Please provide some commentary on whether the use of RT-PCR instead of sequencing could overlook changes since not all bacteria were examined.

Reviewer #2: The manuscript by Nguyen Q, et al, entitled “Alterations in children’s sub-dominant gut microbiota by HIV infection and anti-retroviral therapy” seeks to investigate the gut microbiota of children with HIV in Vietnam and the impact of ART. The paper has important implications as there is no clear consensus on the microbiome in children with HIV, both on and off ART. While the authors find that HIV and ART may influence sub-dominant gut bacteria, there is surprisingly not a significant difference between the microbiome of children uninfected and infected with HIV, opposing current research in the field that has shown reduced bacterial diversity with HIV. Overall, the paper is thought-provoking and provides broad impact; however, there are many key points that the authors need to address:

Major points:

1. A major claim within the paper is that HIV and ART influence the sub-dominant gut microbiota; however, support for this claim rely upon various associations based on the number of bacteria. This case would be strengthened by looking at alpha and beta diversity and the relative abundance of bacterial species.

2. The methods of the RT-qPCR are unclear. The primers appear to be specific for the 14 listed bacteria. If this is true, this eliminates the analysis of a large variety of other intestinal bacteria. Could the authors explain their selection criteria for these 14 selected bacteria? Are these the most abundant bacteria in children? Could the authors also explain their rationale for not sequencing with a universal 16S rRNA primer to identify the most abundant bacteria?

3. FoxP3 and IL-17 are expressed by Tregs and Th17, respectively. Have the authors considered performing a functional analysis of the T cell populations to better classify these T cells?

4. The authors state that they “regarded the threshold for dominance in abundance at 1.0% of the total bacterial count, and the threshold in counts was set at 109 cells/g”. What percentage of the bacterial counts passed this threshold? How was the threshold for sub-dominant bacteria determined at <109 cells/g?

5. Could the authors explain whether the bacterial counts were normalized across groups?

6. It appears that there were many comparisons examined to produce the associations highlighted in this manuscript. Was the data corrected for multiple comparisons using a false discovery rate?

7. Table S2 provides the detection frequency of bacteria in fecal samples from each study group. However, this only accounts for the percentage of samples that harbored the detectable bacteria. Providing the relative abundance of bacterial frequencies within the entire group would strengthen the data and provide a better comparison of the distinct patterns across the groups.

8. The association between number of bacteria and age represent a correlation, not a causation. Can the authors expand upon the impact of the data especially since the groups differed in the average age? Could the positive association be due to the development of the microbiome and the lower age of the HIV(-) group?

9. Can the authors speculate on the impact of the ART duration? Was the duration long enough to see an impact on and restoration of the microbiome?

10. Studies have shown that an altered gut microbiota is associated with elevated circulating inflammatory markers. Since blood was collected in this study, did the authors consider performing an ELISA on the plasma to check for elevated markers of inflammation and microbial translocation?

11. HIV(-) and HIV(+) groups had similar gut microbiota structures both of which differed from the ART(+) group. Could the authors speculate as to whether this was due to ART itself, irrespective of HIV infection?

Minor points:

1. Could the authors include the age at which the children in the ART(+) group started ART?

2. Could the authors provide the specific antiretroviral used in the study, including the specific nucleoside reverse transcriptase inhibitors, non-nucleoside reverse transcriptase inhibitor, and protein inhibitor?

6. PLOS authors have the option to publish the peer review history of their article (what does this mean?). If published, this will include your full peer review and any attached files.

Reviewer #1: **Yes: **Moriah J. Castleman

Reviewer #2: No

---

## [Author Response · Author response to Decision Letter 0]

3 Sep 2021

Responses to the Reviewers' comments:

The authors would like to thank all the reviewers for their valuable suggestions and precise comments to clarify the major contribution of the work. Moreover, we sincerely appreciate the reviewers’ great efforts in pointing out the existing inconsistencies and errors for the improvement. To our best, the manuscript has been carefully revised and clarified according to the reviewers' comments.

Reviewer #1: 

This study examined select bacteria via RT-PCR in the feces of children from Vietnam who were HIV- or HIV+ (ART+/-). The authors then looked for associations between the absolute number of bacteria and a variety of other parameters (Age of child, length of time on ART, T cell counts, activation of CD8+ T cells and use of cotrimoxazole) antibiotic.

Major

Comment 1. This study is similar to previous work including Flygel et al 2020 Journ Infec Dis and Kaur et al 2018 Sci Rep, which the authors did cite in the introduction. This work is also similar to the studies by Dirajlal-Fargo et al 2020 AIDS (Brazil) and Abange et al 2021 Sci Repo (Cameroon) which was not cited by the authors.

A) Since the current study is so similar to other studies, the authors must provide additional justification for the current study in the introduction. 

Response: 

Following the reviewer's comment, we have revised the sentences in the introduction as follows: “Although the composition of the gut microbiota may be influenced by age, diet, genetics, and geography, four phyla (i.e., Firmicutes, Bacteroidetes, Actinobacteria, and Proteobacteria) are dominant and stable in healthy individuals [4,6,7]” (lines 56–58); and “A few studies from Africa and India have shown reduced bacterial diversity in the gut microbiota of children with HIV and children treated with ART compared to the microbiota of children without HIV [23¬¬−25]. However, no consensus exists on whether ART in children with HIV may restore the gut microbiota to the state observed in children without HIV [23¬¬−25].” (lines 74–78)

 The composition of gut microbiota is known to be influenced by geography [ref 7]. Poor understanding of gut microbiota in children with HIV living in different geography is one of the rationales for conducting the current study. Kaur et al. investigated the gut microbiota of HIV-infected and -uninfected children in India, Flygel et al. in Zimbabwe, and Abange et al. in Cameroon. In Vietnam, no study has focused on understanding the gut microbiota in children with HIV so far. 

B) The authors should reference the additional two studies listed above.

Response: 

Following the reviewer's comment, we have included the paper by Abange et al. 2021 Sci Repo as a reference in our revised manuscript as mentioned above. However, we have not included the paper by Dirajlal-Fargo et al. (AIDS, 2020) in the revised manuscript because of the reason as mentioned below:

Dirajlal-Fargo et al. (AIDS, 2020) investigated the correlation between fungal translocation and immune status in the adolescents with and without HIV infection, whose median age was 13 years (IQR 11-15), in Uganda. In contrast, we investigated the composition of gut microbiota and the correlation between gut microbiota and immune status in the children with and without HIV infection, whose age was between 2 and 8 years old, in Vietnam. Thus, their report does not match the context of our current study. Therefore, we have not included the paper in the reference list of this revised manuscript.

 Dirajlal-Fargo et al. (AIDS 2019) also reported that HIV-exposed-uninfected infants had higher levels of inflammation and monocyte activation compared to HIV-unexposed infants at birth, and that the elevated markers of inflammation were associated with a lower weight, in Brazil. As mentioned above, we consider that their report does not match the context of our current study, and have not included their paper in the reference list of the revised manuscript.

 C) There is no mention of these comparable studies in the discussion. The current study’s findings must be discussed in the context of existing literature (i.e. what findings were similar between studies, what findings were different and speculation on why there are differences between studies). For example, in the Flygel et al study, children were on ART for at least 6 months at the time of sampling. How does this differ since it is known that the length of time on ART impacts gut microbiome (Imahashi et al 2021)? How did T cell activation differ between the appropriate studies? Does geographical region impact differences between studies?

Response:

We believe that we have tried to discuss about most of our important findings in the context of existing literatures in each paragraph of the discussion section. However, as we have mentioned in the second paragraph of the discussion section that "our results may not be comparable directly to the results of the other studies using NGS, since the RT-qPCR method is not appropriate to calculate the diversity indices and the relative abundance of the selected bacteria (lines 336−338), we would think that direct comparison of our results with those of other studies conducted in different countries may not be appropriate.

 In addition, we have already mentioned the limitation of our study in the second last paragraph in the discussion section that "First, the children in the HIV(−) group were 2 years younger than the other groups. The diets were not controlled among the groups, though the children in the HIV(−) and ART(+) groups who resided at orphanages were provided the same diets. The children in the HIV(+) group who resided in their own homes appeared to have poorer nutritional status than the children in other groups, which could be due to the uncontrolled diet and/or HIV infection [26]. Considering the influence of age and diet factors on the gut microbiota [7,21,22,33,56–58], we did not focus on comparing the gut microbiota between the groups, but highlighted the factors associated with the gut microbiota in each group." (lines 398-406) 

As for the impacts of the length of time on ART on gut microbiome, we found that ART duration had an independent positive association only with C. perfringens, a sub-dominant gut bacterium, in the HIV-infected children on ART with a median duration of 3.5 years (range: 0.8–5.8 years). This is the most important finding in our study using RT-qPCR, the authors think. The changes of gut microbiota by ART have been reported in HIV-infected children on ART with a minimum duration of 6 months (Flygel, 2020; Imahishi, 2021). On the other hand, Flygel et al. reported that the gut microbiota of children on ART longer than 10 years was similar to that of the HIV-uninfected children. Considering these previous studies, we would think that the duration of ART in our study was long enough to investigate the impact of ART on the gut microbiota, but not long enough to observe the restoration of the gut microbiota by ART.

As for the correlation between gut bacteria and T-cell activation, we could not find any significant correlation between gut bacteria and CD8+-cell activation in the HIV(+) group nor ART(+) group in our study.

Comment 2. The gating strategy in S1 Figure needs some modification.

A) For the CD8 vs CD4 flow plot, most of the events appear to be out of view on the axis. This flow plot needs to be adjusted so that all events are brought into view.

Response:

We have adjusted the gating of CD4+ and CD8+ cells as density plots in Logicle scale to show the events much more clearly.

B) For the CD38 vs HLA-DR flow plot, please provide and isotype control flow plot or FMO controls. The authors may be missing some of the activated CD8 cells based on where the quadrant gate currently sits.

Response:

We did not use isotype control or FMO controls, but used the unstained cells as gating controls. We have added two mini-figures, including no-staining, Per-CP vs. PE, PE-Cy7 vs. FITC, in the S1 Figure. Also, one sentence has been added in the S1 Figure legend as follows: "Unstained cells were used as gating controls.” (line 608)

C) It is confusing why the authors have an arrow pointing downwards from the lymphocyte gate towards CD4 when they could have gated from the CD4+ cells found in the CD8 vs CD4 flow plot.

Response:

Following the reviewer's comment, we have revised the S1 Figure.

Comment 3. The authors need to provide justification for why CD3 was not included in the panel since monocytes and NK cells in the blood can express CD4 and CD8.

Response: 

The data were retrieved from our previous study (ref 26: Bi, et al., 2016). The stained immune cells were analyzed by JSAN flow cytometer (Bay Bioscience, Kobe, Japan) that could detect only 4 colors, which limited us including CD3 in the panel. Before conducting the study, we stained the cells in three colors: CD3, CD4 and CD8, and analyzed them with the flow cytometer to confirm if our cell-gating strategy would be appropriate. We found that CD4 and CD8 molecules were expressed more strongly on the CD3-positive CD4+ and CD8+ cells than CD3-negative CD4+ and CD8+ cells, respectively (below figures). We, therefore, gated the cells with highly-expressed CD4 and CD8 molecules as CD4+ and CD8+ cells, respectively, as shown in the below figure C and S1 Figure. The cells with highly-expressed CD4 molecule (46.17% in the figure C) were almost equivalent to the CD3+CD4+ cells (45.58% in the figure A), and the cells with highly-expressed CD8 molecule (17.51% in the figure C) to the CD3+CD8+ cells (17.86% in the figure B). In addition, as monocytes are larger than lymphocytes, most of the monocytes could be gated out by our lymphocyte gating strategy, especially when we used fresh whole blood for staining (S1 Figure). Thus, we thought that our gating strategy could be used for this study.

Minor

Comment 1. Provide some explanation in the discussion as to why the antibiotic cotrimoxazole only effected C. perfringens.

Response: 

We have already discussed some about cotrimoxaole in the 6th paragraph in the discussion section as follows: "The use of cotrimoxazole reportedly influences some gut bacteria and reduces gut inflammation in children with HIV [48−50]. In the current study, the use of cotrimoxazole was associated only with C. perfringens in the ART(+) group. However, in multiple regression analysis, we found that ART duration, but not the use of cotrimoxazole, was independently associated with C. perfringens, which is a potentially harmful bacterium [51]."(lines 379−384). Thus, we determined that the association of the cotrimoxazole with C. perfringens in the ART(+) group was a spurious correlation.

Comment 2. Please provide some commentary on whether the use of RT-PCR instead of sequencing could overlook changes since not all bacteria were examined.

Response: 

We have already discussed our choice of an RT-qPCR approach rather than NGS in the second paragraph in the discussion section in the original manuscript. Following the reviewer's comment, we have added some limitation of the RT-qPCR approach in the second paragraph in the discussion section (lines 336–338).

Reviewer #2: The manuscript by Nguyen Q, et al, entitled “Alterations in children’s sub-dominant gut microbiota by HIV infection and anti-retroviral therapy” seeks to investigate the gut microbiota of children with HIV in Vietnam and the impact of ART. The paper has important implications as there is no clear consensus on the microbiome in children with HIV, both on and off ART. While the authors find that HIV and ART may influence sub-dominant gut bacteria, there is surprisingly not a significant difference between the microbiome of children uninfected and infected with HIV, opposing current research in the field that has shown reduced bacterial diversity with HIV. Overall, the paper is thought-provoking and provides broad impact; however, there are many key points that the authors need to address:

Major points: 

Comment 1. A major claim within the paper is that HIV and ART influence the sub-dominant gut microbiota; however, support for this claim rely upon various associations based on the number of bacteria. This case would be strengthened by looking at alpha and beta diversity and the relative abundance of bacterial species.

Response: 

In this study, we did not employ the sequencing approach, but used RT-qPCR approach for analyzing specific bacteria. Because of large range of the bacteria (10^2-10^10 cells/g feces), we thought it would not be appropriate to calculate the diversity indices and the relative abundance from the microbiota data of the selected bacteria. This discussion has been added in the second paragraph of the discussion section as follows: "though our results may not be comparable directly to the results of the other studies using NGS, since the RT-qPCR method is not appropriate to calculate the diversity indices and the relative abundance of the selected bacteria. (lines 336–338) 

Comment 2. The methods of the RT-qPCR are unclear. The primers appear to be specific for the 14 listed bacteria. If this is true, this eliminates the analysis of a large variety of other intestinal bacteria. Could the authors explain their selection criteria for these 14 selected bacteria? Are these the most abundant bacteria in children? Could the authors also explain their rationale for not sequencing with a universal 16S rRNA primer to identify the most abundant bacteria?

Response: 

As the reviewer pointed out, we used the specific primer sets for the 14 listed bacteria. We selected Clostridium coccoides group, Clostridium leptum subgroup, Bacteroides fragilis group, Bifidobacterium, Atopobium cluster, and Prevotella, since more than 70% of total intestinal bacteria were covered by these groups [ref 29]. The other 8 bacterial groups including lactobacilli and potential pathogens were selected from the perspectives of their associations with health and diseases. We employed the RT-qPCR approach since it is proven to be an efficient and valuable tool for an exhaustive analysis of gut microbiota over a wide dynamic range. The RT-qPCR approach can detect and enumerate the gut bacteria at the population level between 102 and 1011 cells/g of stool, while the lower detection limit of the sequencing approach is 107 to 108 cells/g of feces. The counts of the subdominant bacteria were around the detection limit of the sequencing approach or lower [ref 33], we thus took advantage of RT-qPCR to estimate the counts of these less abundant but clinically significant targets. 

Comment 3. FoxP3 and IL-17 are expressed by Tregs and Th17, respectively. Have the authors considered performing a functional analysis of the T cell populations to better classify these T cells?

Response:

The immunological data were retrieved from our previous study (ref 26: Bi, et al, 2016). It is obvious that the cytokine production is the best marker for classifying CD4 cell subsets such as Th17. Unfortunately, we could not carry out cell stimulation experiment at that time mainly due to the limited amount of the blood samples collected from the children with HIV infection (2-3ml of whole blood per child). Thus, we used cell surface markers to define the cells as an alternative method (ref 26: Bi, et al, 2016). 

Comment 4. The authors state that they “regarded the threshold for dominance in abundance at 1.0% of the total bacterial count, and the threshold in counts was set at 109 cells/g”. What percentage of the bacterial counts passed this threshold? How was the threshold for sub-dominant bacteria determined at <109 cells/g?

Response: 

The previous reports revealed that the average of total bacterial counts was around 1011 cells/g of feces [ref 29,31]. We regarded the threshold for dominance in abundance at 1.0% of total bacterial counts, and thus the threshold in counts was set at 109 cells/g [ref 6,33–35]. We have already mentioned quantification of bacteria in human stool with RT-qPCR paragraph in the methodology section: “Previous reports revealed that the average total bacterial count is approximately 1011 cells/g of feces [29,31]. We regarded the threshold for dominance in abundance at 1.0% of the total bacterial count, and the threshold in counts was set at 109 cells/g [6,33−35].” (lines 133–136). In detail, the dominant bacteria in the gut microbiota (≥109 cells/g of feces) included C. coccoides group, C. leptum subgroup, Bifidobacterium, Atopobium cluster, B. fragilis group, and Prevotella. The sub-dominant gut microbiota (<109 cells/g) included C. difficile, C. perfringens, Streptococcus, Enterobacteriaceae, Lactobacillus spp., Enterococcus, Staphylococcus, and Pseudomonas. In this study, the sum of the dominant gut microbiota accounted for 90.14% ± 14.66% (mean ± SD) of the total gut microbiota, and the percentage of the sub-dominant gut microbiota was 9.86%.

Comment 5. Could the authors explain whether the bacterial counts were normalized across groups?

Response: 

We have already mentioned in the methods parts: “In the same experiment, a standard curve was generated with the RT-qPCR data (by threshold cycle: CT value) and the cell counts (by DAPI staining) of the dilution series of total RNA from the standard strain for each bacterial target. The CT values from fecal samples were normalized to the standard curve to obtain the bacterial count per gram wet weight of feces.” (lines 126–130)

The processing of stool samples was standardized by those wet weights. Fecal samples were weighed, and their portion of 4 mg was subjected to RNA extraction as written in the method part (lines 114–130). In RT-qPCR assay, a standard curve was generated from dilution series of total RNA extracted from the standard strain for each bacterial target based on the cell counts, and the RT-qPCR data of fecal samples were normalized to the standard curve to obtain the bacterial count per gram wet weight of feces of all the children in the same batch.

Comment 6. It appears that there were many comparisons examined to produce the associations highlighted in this manuscript. Was the data corrected for multiple comparisons using a false discovery rate?

Response: 

As we have mentioned in the study limitation paragraph in the discussion section, "Considering the influence of age and diet factors on the gut microbiota [7,21,22,33,56–58], we did not focus on comparing the gut microbiota between the groups, but highlighted the factors associated with the gut microbiota in each group." (lines 403–406). Additionally, to avoid type 1 error (false positive), we conducted a simple correlation analysis and confirmed with multiple linear analysis.

Comment 7. Table S2 provides the detection frequency of bacteria in fecal samples from each study group. However, this only accounts for the percentage of samples that harbored the detectable bacteria. Providing the relative abundance of bacterial frequencies within the entire group would strengthen the data and provide a better comparison of the distinct patterns across the groups.

Response: 

In the current study, we quantified the dominant bacteria in the gut microbiota (≥109 cells/g of feces), including C. coccoides group, C. leptum subgroup, Bifidobacterium, Atopobium cluster, B. fragilis group, and Prevotella. The sub-dominant gut microbiota (<109 cells/g) included C. difficile, C. perfringens, Streptococcus, Enterobacteriaceae, Lactobacillus spp., Enterococcus, Staphylococcus, and Pseudomonas. There was a big gap of the percentage of relative abundance between dominant groups and subdominant groups. Taking advantage of RT-PCR method, we could detect absolute number of bacteria, and focused on the analysis of the factors associated with the gut microbiota in each group.

 C. difficile and Pseudomonas were not included in further analyses due to the low detection frequencies (3.4% to 20% in all three groups). The other bacteria groups with the detection frequency of greater than 50% were added dummy data for further analyses. Missing values were imputed using the half of dectected limitation values. 

Comment 8. The association between number of bacteria and age represent a correlation, not a causation. Can the authors expand upon the impact of the data especially since the groups differed in the average age? Could the positive association be due to the development of the microbiome and the lower age of the HIV(-) group?

Response: 

As the reviewer pointed out, the average age of the children in the HIV(–) group was 2 years younger than the other groups. This would limit our data comparison among the groups. Considering these, we have tried to make it clear that we did not focus on comparing the gut microbiota between the groups but highlighted the analysis of the factors associated with the gut microbiota in each group. Please refer to the second last paragraph of the discussion section (lines 398–406): “First, the children in the HIV(−) group were 2 years younger than the other groups. The diets were not controlled among the groups, though the children in the HIV(−) and ART(+) groups who resided at orphanages were provided the same diets. The children in the HIV(+) group who resided in their own homes appeared to have poorer nutritional status than the children in other groups, which could be due to the uncontrolled diet and/or HIV infection [26]. Considering the influence of age and diet factors on the gut microbiota [7,21,22,33,56–58], we did not focus on comparing the gut microbiota between the groups, but highlighted the factors associated with the gut microbiota in each group.” 

Comment 9. Can the authors speculate on the impact of the ART duration? Was the duration long enough to see an impact on and restoration of the microbiome?

Response: 

Our multiple linear regression analysis showed a significant association only between C. perfringens and ART duration. The HIV-infected children in our study were on ART with a median duration of 3.5 years [range: 0.8–5.8]. The changes of gut microbiota by ART have been reported in HIV-infected children on ART with a minimum duration of 6 months (Flygel, 2020; Imahishi, 2021). On the other hand, Flygel et al. reported that the gut microbiota of children on ART longer than 10 years was similar to that of the HIV-uninfected children. Considering these previous studies, we would think that the duration of ART in our study was long enough to investigate the impact of ART on the gut microbiota, but not long enough to observe the restoration of the gut microbiota by ART. 

Comment 10. Studies have shown that an altered gut microbiota is associated with elevated circulating inflammatory markers. Since blood was collected in this study, did the authors consider performing an ELISA on the plasma to check for elevated markers of inflammation and microbial translocation?

Response: 

In our previous study targeting same cohort (ref 26: Bi, at al, 2016), we could not detect any bacterial 16S/23S rRNA from all children, though a few of the targeted bacterial 16S/23S rRNA gene (rDNA) were detected in the children of HIV(+) and HIV (–) groups but not in the ART(+) group. The level of sCD14 was not significantly associated with the detection frequency of the bacterial rDNA in the serum. Therefore, we did not include sCD14 as an indicator of microbial translocation in the current analysis. We also measured IL-2, 4, 6, 10, 17, IFN-γ, and TNF-α in the serum using two different Multiplex cytokine detection kits, however, all of these cytokines but IFN-γ were under the detection limits. Therefore, we did not include these data in current study. Other makers for inflammation and microbial translocation such as sTNFR-I and II, sCD163, IP-10, D-Dimer, hsCRP and so on, could not be evaluated due to the limited amount of the blood samples collected from the children (2-3ml of whole blood per child).

Comment 11. HIV(-) and HIV(+) groups had similar gut microbiota structures both of which differed from the ART(+) group. Could the authors speculate as to whether this was due to ART itself, irrespective of HIV infection?

Response: 

As we mentioned in the study limitation paragraph, the children in the HIV(−) group were 2 years younger than the other groups and the diets were not controlled among the groups, though the children in the HIV(−) and ART(+) groups who resided at orphanages were provided the same diets. Considering the influence of age and diet factors on the gut microbiota, we did not focus on comparing the gut microbiota between the groups but highlighted the factors associated with the gut microbiota in each group. 

 As we have mentioned in the first paragraph of the discussion section, "We found that several sub-dominant gut bacteria were positively associated with age in children without HIV, but this was not observed in the children with HIV. In addition, Staphylococcus negatively correlated with age, i.e. the duration of HIV infection, in the children vertically infected with HIV, and ART duration had an independent positive association with C. perfringens, a sub-dominant gut bacteria, in the HIV-infected children on ART. These findings indicate an impact of HIV infection and ART on the sub-dominant gut microbiota in children." (lines 319-326) Thus, we would think that HIV infection and ART influence the sub-dominant gut microbiota, directly or indirectly, in association with the immune status of children with HIV. (lines 45-46)

Minor points:

Comment 1. Could the authors include the age at which the children in the ART(+) group started ART?

Response: 

Children in the ART(+) group started ART with the median age of 2.67 (0.42–6.92) years. This information has been added in the Table 1.

Comment 2. Could the authors provide the specific antiretroviral used in the study, including the specific nucleoside reverse transcriptase inhibitors, non-nucleoside reverse transcriptase inhibitor, and protein inhibitor?

Response: 

Following the reviewer's comment, the following sentence has been added in the results section (lines 205–210): "8 received zidovudine/lamivudine/nevirapine; 7 received stavudine/lamivudine/nevirapine; 6 received zidovudine/lamivudine/efavirenz; 4 received stavudine/lamivudine/efavirenz; 2 received zidovudine/lamivudine/lopinavir boosted with ritonavir; 1 received abacavir/lamivudine/lopinavir boosted with ritonavir; and 1 received abacavir/didanosine/lopinavir boosted with ritonavir." 

 

Editorial comment 3. We note that you have included the phrase “data not shown” in your manuscript. Unfortunately, this does not meet our data sharing requirements. PLOS does not permit references to inaccessible data. We require that authors provide all relevant data within the paper, Supporting Information files, or in an acceptable, public repository. Please add a citation to support this phrase or upload the data that corresponds with these findings to a stable repository (such as Figshare or Dryad) and provide and URLs, DOIs, or accession numbers that may be used to access these data. Or, if the data are not a core part of the research being presented in your study, we ask that you remove the phrase that refers to these data.

Response: 

We have re-conducted the multiple linear regression analysis for Staphylococcus and found that the Th17 count was independently associated with the number of Staphylococcus in the ART(+) group. We sincerely apology our mistake.

However, fortunately the change did not affect our conclusion: “HIV infection and ART may influence sub-dominant gut bacteria, directly or indirectly, in association with immune status in children with HIV”, since Staphylococcus belongs to the sub-dominant gut microbiota.

The sentences have been added or revised in the abstract, results, and discussion section as follows: 

In the abstract: 

“In multiple linear regression analyses, ART duration was independently associated with C. perfringens, and and Th17 cell count with the number of Staphylococcus in the ART(+) group.” (lines 42–44)

In the result section: 

”The multiple linear regression analyses including age, ART duration, immune status, and use of cotrimoxazole showed that the ART duration was independently associated with the number of C. perfringens (Beta coefficient = 0.726, P < 0.001) and the Th17 count with the number of Staphylococcus (Beta coefficient = 0.428, P = 0.02) in the ART(+) group (Tables 2 and 3). The linear regression analysis for C. coccoides group in the HIV(+) group showed no significant association (S5 Table).” (lines 299–304)

In the discussion part: 

“We found that several sub-dominant gut bacteria were positively associated with age in children without HIV, but this was not observed in the children with HIV. In addition, Staphylococcus negatively correlated with age, i.e. the duration of HIV infection, in the children vertically infected with HIV, and ART duration had an independent positive association with C. perfringens and Th17 count with Staphylococcus in the HIV-infected children on ART. These findings indicate an impact of HIV infection and ART on the sub-dominant gut microbiota, including C. perfringens and Staphylococcus, in children.” (lines 311–318)

Fifth paragraph has been newly added as follows:

“Multiple regression analysis showed a positive association between gut Staphylococcus and Th17 counts in the ART(+) group, which was shown for the first time. Th17 cells produce interleukin-17, which is important for promoting neutrophil recruitment to clear bacteria and has a specific role in the host defense against Staphylococcus aureus skin infection [47]. Thus, it would be interesting to investigate the interaction between Th17 and gut Staphylococcus in order to understand the pathophysiology of HIV infection in children who are on ART.” (lines 361–367)

The title of the table 2 has been modified as follows: Table 2. Linear regression analysis of Clostridium perfringens with age, ART duration, immune status, and use of cotrimoxazole in the ART(+) group

Table 3 and S5 Table have been newly added in the revised manuscript.

---

## [Decision Letter · Decision Letter 1]

22 Sep 2021

Alterations in children's sub-dominant gut microbiota by HIV infection and anti-retroviral therapy

PONE-D-21-16737R1

Dear Dr. Ichimura,

We’re pleased to inform you that your manuscript has been judged scientifically suitable for publication and will be formally accepted for publication once it meets all outstanding technical requirements.

Kind regards,

Jennifer Manuzak

Academic Editor

PLOS ONE

Additional Editor Comments (optional):

Reviewers' comments:

Reviewer's Responses to Questions

**Comments to the Author**

1. If the authors have adequately addressed your comments raised in a previous round of review and you feel that this manuscript is now acceptable for publication, you may indicate that here to bypass the “Comments to the Author” section, enter your conflict of interest statement in the “Confidential to Editor” section, and submit your "Accept" recommendation.

Reviewer #1: All comments have been addressed

Reviewer #2: All comments have been addressed

2. Is the manuscript technically sound, and do the data support the conclusions?

Reviewer #1: Partly

Reviewer #2: (No Response)

3. Has the statistical analysis been performed appropriately and rigorously? 

Reviewer #1: Yes

Reviewer #2: (No Response)

4. Have the authors made all data underlying the findings in their manuscript fully available?

Reviewer #1: Yes

Reviewer #2: (No Response)

5. Is the manuscript presented in an intelligible fashion and written in standard English?

Reviewer #1: Yes

Reviewer #2: (No Response)

6. Review Comments to the Author

Reviewer #1: (No Response)

Reviewer #2: (No Response)

7. PLOS authors have the option to publish the peer review history of their article (what does this mean?). If published, this will include your full peer review and any attached files.

Reviewer #1: No

Reviewer #2: No

---

## [Editor Report · Acceptance letter]

30 Sep 2021

PONE-D-21-16737R1 

Alterations in children's sub-dominant gut microbiota by HIV infection and anti-retroviral therapy 

Dear Dr. Ichimura:

I'm pleased to inform you that your manuscript has been deemed suitable for publication in PLOS ONE. Congratulations! Your manuscript is now with our production department. 

Kind regards, 

on behalf of

Dr. Jennifer Manuzak 

Academic Editor

PLOS ONE